# The Mechanism of Lipopolysaccharide’s Effect on Secretion of Endometrial Mucins in Female Mice during Pregnancy

**DOI:** 10.3390/ijms23179972

**Published:** 2022-09-01

**Authors:** Dezhang Lu, Wenxiang Hu, Tian Tian, Mengran Wang, Mengru Zhou, Chenchen Wu

**Affiliations:** College of Animal Veterinary Medicine, Northwest A&F University, Yangling District, Xianyang 712100, China

**Keywords:** lipopolysaccharide, uterus, mucous layer, PI3K/AKT/NF-κB signaling pathway, tight junction protein

## Abstract

The main toxic component of endotoxins released from the death or dissolution of Gram-negative bacteria is lipopolysaccharide (LPS), which exists widely in the natural environment, and a large amount of endotoxin can significantly inhibit the reproductive performance of animals. A previous study showed that endotoxins mainly damaged the physiological function of mucins in the endometrium, but the mechanism is not clear. In this study, the PI3K/Akt signaling pathway was not activated, and the NF-κB signaling pathway was inhibited by LPS treatment; the expression of occludin and E-cadherin proteins were decreased and ZO-1 protein expression was increased, because LPS can lead to the mucous layer becoming thinner, so that the embryonic survival rate is significantly reduced in early pregnancy. In middle and late pregnancy, LPS translocated to the epithelial cells of the uterus and the expression of claudin-1, JAMA, and E-cadherin proteins were decreased; at this time, a large number of glycosaminoglycan particles were secreted by endometrial gland cells through the PI3K/Akt/NF-κB signaling pathway that was activated after LPS treatment, However, there was no significant difference between the survival rates of fetal mice in the LPS (+) and LPS (-) groups. Glycosaminoglycan particles and mucins are secreted by gland cells, which can protect and maintain the pregnancy in the middle and late gestational periods.

## 1. Introduction

Bacterial endotoxins are cell wall components of Gram-negative bacteria, which are released when bacterial death or autolysis occurs, and their main component is lipopolysaccharide (LPS) [1]. LPS is the major virulence factor in pathogenic Gram-negative bacteria and covers nearly 75% of the cell surface [2,3]. Bacterial endotoxins are widely found in the environment and food. When a small amount of endotoxin enters the human or animal body through the digestive tract, it does not cause damage; however, when a large amount of endotoxin enters the blood, it can cause different diseases [4,5]. Previous studies have shown endotoxin changes in uterine histomorphology in mice, particularly in the function of the mucous layer of the endometrium [6]. Mucins are large, highly O-glycosylated proteins that protect epithelial surfaces. Mucins are divided into two, categories based on their functions, membrane and gel-forming. Gel-forming mucins form mucous gels, which are complex viscoelastic gels that form on the surface of the secretory epithelium and act as the first line of defense against harmful substances from the outside environment [7,8,9]. The gel-forming mucins provide hydration, lubrication, transport, and protection to the mucosa during pregnancy [10]. They are primarily produced by specialized cells, such as goblet cells embedded in epithelia or gland cells. Membrane mucins not only provide the ultimate barrier for the epithelial surface, but also stimulate additional protective mechanisms, including promoting cell survival through their involvement in cell signaling [11,12]. The purpose of this study was to provide comprehensive research on lipopolysaccharide (LPS) and the mucous layer of the endometrium and to describe the mechanism of the relationship between LPS and mucins in the uterus during pregnancy and childbirth. This could be the foundation for future research on reproductive disorders caused by a large number of endotoxins in the working environment.

## 2. Results

### 2.1. Effect of LPS on the Level of Endometrial Mucins in Mice

To confirm the effect of LPS on mucin (MUC1, MUC2, MUC3, MUC4, MUC5ac, MUC5b, MUC6, MUC13, and MUC20) levels in the endometrium, 20 female mice were intragastrically administered LPS reagent every other day for one month. The results showed that MUC1, MUC3, and MUC5b mRNA expressions in the endometrium did not differ between the LPS (+) and LPS (-) groups (*p* > 0.05). MUC5ac and MUC20 mRNA expressions in the endometrium were very low in the LPS (+) and LPS (-) groups. MUC2 mRNA expression in the endometrium was significantly increased in the LPS (-) group, compared with that in the LPS (+) group (*p* < 0.05). MUC4 mRNA expression in the endometrium was significantly decreased in the LPS (-) group, compared with that in the LPS (+) group (*p* < 0.05) (Figure 1A).

Furthermore, we used ELISA to analyze MUC2 and MUC4 protein content because there was a significant difference in mRNA expression between the LPS (+) and LPS (-) groups. MUC2 protein content in the endometrium of mice in the LPS (-) group was higher than that in mice in the LPS (+) group (*p* < 0.05). MUC4 protein content of the endometrium in mice in the LPS (-) group was not different from that of mice in the LPS (+) group (*p* > 0.05). These results indicate that LPS has a significant effect on MUC2 and MUC4 content in the endometrium of mice (Figure 1B).

### 2.2. Effect of LPS on MUC2 and MUC4 Expression in the Endometria of Mice during the Pregnancy and Childbirth Periods

In Section 2.1, we noted that LPS can significantly affect MUC2 and MUC4 expression; therefore, we wanted to determine how LPS affects MUC2 and MUC4 expression in the endometrium during pregnancy and childbirth. At day 7 of pregnancy, MUC2 expression was obviously increased in the LPS (-) group. MUC2 expression in the endometrium of mice in the LPS (+) group was higher than that in the LPS (-) group at day 15 of pregnancy and 5 days after childbirth (Figure 2A).

At day 7 of pregnancy, MUC4 expression was obviously increased in the LPS (-) group. At day 15 of pregnancy, MUC4 expression in the endometrium of the LPS (+) group was higher than that in the LPS (-) group. Five days after childbirth, MUC4 expression in the endometrium of the LPS (-) group was not different from that in the LPS (+) group (Figure 2B).

To confirm the above results, we measured MUC2 and MUC4 levels in the endometrium of the LPS (-) and LPS (+) groups by ELISA. At day 7 of pregnancy, MUC2 and MUC4 contents in the endometrium were significantly increased in the LPS (-) group, compared to the LPS (+) group (*p* < 0.05). At day 15 of pregnancy, the endometrial contents of MUC2 and MUC4 in the LPS (-) group were lower than those in the LPS (+) group (*p* < 0.05). Five days after childbirth, the endometrial contents of MUC2 and MUC4 in the LPS (-) group were not different from those in the LPS (+) group (*p* > 0.05) (Figure 2C).

### 2.3. Effect of LPS on Histomorphology and Ultrastructure of the Uterus in Mice

Mucopolysaccharides were determined using periodic acid–Schiff (PAS) staining. At day 7 of pregnancy, the results found that the mucous layer thickness was increased in the LPS (-) group (black arrow), and the mucous layer of the endometrium in the LPS (+) group was very thin after LPS stimulation. At day 15 of pregnancy, the mucous layer of the endometrium in the LPS (-) group became very thin. The mucous layer of the endometrium in the LPS (+) group was still very thin; however, the endometrial glandular cells of the LPS (+) group secreted a large number of glycosaminoglycan particles (yellow arrow). At 5 days after childbirth, the endometrial glandular cells of the LPS (+) group secreted many glycosaminoglycan particles (green arrow); however, the endometrial glandular cells of the LPS (-) group did not secrete many glycosaminoglycan particles (Figure 3A).

Ultrastructures of the uterine glandular epithelial cells and cytoplasmic polysaccharide granules were analyzed. The endometrial glandular cells secreted more glycosaminoglycan particles at day 15 of pregnancy than at 5 days after childbirth in the LPS (+) group (red box) (Figure 3B). These combined results indicate a thicker mucous layer in the endometrium during the early pregnancy, which becomes thinner during the middle and late pregnancy periods under normal conditions. However, the higher the LPS dose, the more glycosaminoglycan particles were secreted by glandular cells. At 5 days after childbirth, the number of glycosaminoglycan particles decreased in the LPS (+) group.

### 2.4. Effect of LPS on Tight Junction Protein of Endometrial Epithelial Cells

We measured claudin-1, JAMA, occludin, E-cadherin, and ZO-1 protein expressions using western blot (Figure 4). At day 7 of pregnancy, endometrial claudin-1 and JAMA protein expressions in the LPS (+) group were not different from those in the LPS (-) group (*p* > 0.05). Endometrial occludin and E-Cadherin protein expressions in the LPS (+) group were decreased, compared with those in the LPS (-) group (*p* < 0.05). Endometrial ZO-1 protein expression in the LPS (+) group was increased, compared with that in the LPS (-) group (*p* < 0.05).

At day 15 of pregnancy, endometrial ZO-1 protein expression in the LPS (+) group was not different from that in the LPS (-) group (*p* > 0.05). Endometrial claudin-1, JAMA, and E-cadherin protein expressions in the LPS (+) group were decreased, compared with those in the LPS (-) group (*p* < 0.05). Endometrial occludin protein expression in the LPS (+) group was increased, compared with that in the LPS (-) group (*p* < 0.05). At day 5 after of childbirth, there were no differences in ZO-1, claudin-1, JAMA, E-cadherin, or occludin protein expressions in the endometrium between the LPS (+) and LPS (-) groups (*p* > 0.05).

### 2.5. Effect of LPS on Reproductive Performance of Mice

The fetal mouse survival rate at day 7 of pregnancy and day 5 after childbirth in the LPS (-) group was increased compared with that in the LPS (+) group (*p* < 0.05). There was no difference in the fetal mouse survival rate at day 15 of pregnancy between the LPS (+) and LPS (-) groups (*p* > 0.05) (Figure 5).

### 2.6. Effect of LPS on PI3K/AKT/NF-κB Signaling Pathway in the Pregnancy Period

This study aimed to provide insights into the mechanism by which LPS affects mouse endometrial tissue. A previous study found that gland cells secrete glycosaminoglycan particles into the uterine cavity, which is regulated by the PI3K/AKT/NF-κB signaling pathways [13]. Therefore, in this study, we analyzed the mRNA expression of the PI3K/AKT/NF-κB signaling pathway. Altogether, the above results indicate that LPS can significantly influence the function of the endometrium of mice at days 7 and 15 of pregnancy; however, LPS did not affect the function of the endometrium of mice at 5 days after childbirth. Therefore, we only measured the mRNA expression of the PI3K/AKT/NF-κB signaling pathway at days 7 and 15 of pregnancy.

At day 7 of pregnancy, there were no differences in PI3K, AKT, PDK1, TAK1, IKKα, IKKβ, an IκBα mRNA expressions in the endometria of mice between the LPS (-) and LPS (+) groups (*p* > 0.05). NF-κB p50, NF-κB p65, IL-6, and TNF-α mRNA expressions in the endometrium were significantly increased in the LPS (-) group, compared to those in the LPS (+) group (*p* < 0.05). IL-1β mRNA expression in the endometrium was significantly decreased in the LPS (-) group, compared to that in the LPS (+) group (*p* < 0.05) (Figure 6A).

At day 15 of pregnancy, there were no differences in PI3K, IKKβ, and NF-κB p50 mRNA expression in the endometrial tissue of mice in the LPS (-) and LPS (+) groups (*p* > 0.05). AKT, PDK1, TAK1, IKKα, IκBα, IL-1β, NF-κB p65, IL-6, and TNF-α mRNA expressions in the endometrial tissue of mice were significantly increased in the LPS (+) group, compared to those in the LPS (-) group (*p* < 0.05) (Figure 6B).

## 3. Discussion

Bacterial endotoxins are produced by Gram-negative bacteria in the cell. The toxins released from broken and dead cells are heat-resistant and stable [14], and their main toxic component is lipopolysaccharide (LPS). Since Gram-negative bacteria are ubiquitous in the environment, LPS is also ubiquitous in the environment [15]. When a large amount of endotoxin enters humans and animals, it can cause various lesions [16]. Previous experimental studies have showed that LPS levels in the uterus were significantly higher than those in the ileum and colon on the 35th day after LPS treatment, and the rate of pregnancy and survival were significantly decreased in the LPS (+) group compared to the LPS (-) group [17]. These results indicate that reproductive dysfunction caused by LPS mainly damages the normal physiological function of the uterus. The uterus is the organ involved in pregnancy, it receives the fertilized egg and allows it to develop [18]. The uterus is made up of a thick, muscular myometrium, a spongy mucosal layer, and an endometrium that provides shelter and support for the developing embryo and fetus during pregnancy [19]. The mucus layer covering the uterine epithelial cells is composed of macromolecular glycoprotein mucins [20,21], which function is to offer protection against pathogens and other toxic factors [22,23,24]. This study primarily focused on quantifying the mRNA expression of nine mucin genes (MUC1, MUC2, MUC3, MUC4, MUC5AC, MUC5B, MUC6, MUC13, and MUC20) in the uterus of mice in LPS (+) and LPS (-) groups. The MUC2 and MUC4 mRNA expression were higher than other mucins in the endometrium, and were significantly different between the LPS (+) and LPS (-) groups. Combined immunofluorescence and PAS staining results have showed that MUC2 and MUC4 are secreted proteins produced by epithelial goblet cells as the main component of the mucus layer in the endometrium [19]. Under normal conditions, the thickness of the mucous layer in the uterus is beneficial for fetal implantation during early pregnancy, but the uterine mucous layer becomes thinner in middle or late pregnancy. After LPS treatment, the mucous layer of the uterus became thinner during early pregnancy, and the uterine gland cells secreted a large number of glycosaminoglycan particles in middle or late pregnancy, which were released into the uterine cavity by exocytosis. As well as the results on the survival rate of fetal mice, the number of embryos decreased significantly after the mucous layer was thinned by LPS treatment at day 7 of pregnancy. The number of embryos was not significantly different between the LPS (+) and LPS (-) groups at day 15 of pregnancy, indicating that the glycosaminoglycan particles secreted by uterine gland cells have a good protective effect on the physiological function of the uterus and play an important role in maintaining the pregnancy.

LPS damaged the structure of the mucous layer of the endometrium and translocated to endometrial epithelial cells. We further analyzed the changes in tight junction protein expression between the epithelial cells of the uterine tissue. The tight junctions of the endometrial epithelial cells below the mucous layer function to close the gap between cells, form an epithelial cell barrier, and participate in the formation of the uterine epithelial mechanical barrier (Figure 4F). At day 7 of pregnancy, LPS caused almost no damage to the mucous layer structure of the endometrium, when occludin protein expression was significantly decreased, and ZO-1 protein expression in the LPS (+) group was significantly increased compared with that in the LPS (-) group. The uterus up-regulated the expression of ZO-1 to adapt to LPS damage to tight junctions. At day 15 of pregnancy, LPS damaged the mucous layer structure of the endometrium, and it can reduce the expression of claudin-1 and JAMA proteins. JAMA, claudin-1, and occludins together form a tight junction transmembrane structure [25]. LPS significantly reduced E-cadherin expression throughout pregnancy. E-cadherin is the most important member of the cadherin family involved in cell adhesion and plays an extremely important role in the adhesion between cells and in maintaining the morphological structure of tissues [26,27]. In the uterus, lack of E-cadherin expression is an important cause of endometriosis [28,29]. LPS may interfere with the cell–cell adhesion of endometrial epithelial cells by affecting E-cadherin, thereby destroying the epithelial cell mechanical barrier. These results suggest that LPS can increase the thickness of endometrial mucous layer, and it does not damage cell–cell tight junctions due to the protection of the mucus layer in early pregnancy; LPS reduces the thickness of the endometrial mucus layer, and the expression of tight junction proteins significantly decreases, which destroys the cell–cell tight junctions in middle and late pregnancy. At day 5 after childbirth, LPS had no significant effect on mucin and tight junction proteins in the endometrium. At this time, the lactation function of the female mice was hyperactive, and LPS in the blood might have accumulated in the mammary glands during lactation and been absorbed by the offspring. There may be a toxic effect of LPS on the offspring through breast milk, but further research is required to ascertain this.

LPS destroyed the endometrial mucous layer and translocated to the uterine epithelial cells. The above study results show that LPS promotes the secretion of glycosaminoglycan particles by endometrial gland cells to maintain pregnancy function after destroying the structure of the mucous layer. However, we do not know the mechanism of LPS regulation on mucin secretion. The gland cells secrete glycosaminoglycan particles into the uterine cavity by exocytosis to increase the thickness of the mucous layer to protect the uterus, and previous studies have shown that this process is regulated by the PI3K/AKT/NF-κB signaling pathways [30,31,32]. At day 7 of pregnancy, NF-κB p50 and NF-κB p65 mRNA expressions in the endometrial tissue were significantly decreased in the LPS (+) group, compared to those in the LPS (-) group. This indicated that the PI3K/Akt signaling pathway was not activated, and the NF-κB signaling pathway was inhibited in early pregnancy. At day 15 of pregnancy, Akt, PDK1, IKKα, and TAK1 mRNA expressions in the endometrial tissue were significantly higher in the LPS (+) group than in the LPS (-) group. The PI3K/Akt signaling pathway was activated after LPS treatment. However, IκBα and NF-κB p65 mRNA expressions in the endometrial tissue were significantly increased in the LPS (+) group, compared with those in the LPS (-) group in the downstream pathway, indicating that the NF-κB signaling pathway was also activated.

The heterodimer of NF-κB is composed of p65, p50, and IκBα subunits, which exist in the cytoplasm in an inactive state. In response to to LPS stimulation, the IκBα subunit is phosphorylated and degraded, thereby promoting the translocation of the p50-p65 heterodimer to the nucleus, where the p50-p65 heterodimer acts as a transcription factor regulating mucin expression [33,34]. NF-κB activation involves phosphorylation of IκBα by IKKs (IKKα and IKKβ), resulting in IκBα degradation. Consequently, NF-κB is released and translocates freely into the nucleus. Mucin acts on airway epithelial cells by regulating the NF-κB signaling pathway [35]. Downstream activation of the PI3K/Akt/NF-κB signaling pathway can lead to increased levels of IL-1β, IL-6, and TNF-α in the uterus. IL-1β induces MUC2 gene expression and mucin secretion through activation of the PKC-MEK/ERK-dependent pathway and PI3K [36]. IL-1β can activate PKC-MEK/ERK and PI3K signaling pathways, inducing MUC2 gene expression and mucin secretion. Meanwhile, PI3K also plays an important role in the up-regulation of MUC2 expression and mucin secretion induced by IL-1β. TNF-α has been reported to be associated with mucin secretion [37,38]. Our results demonstrate that LPS inhibits the PI3K/Akt/NF-κB signaling pathway, and that the structure of the mucous layer of the endometrium is damaged during early pregnancy. LPS translocates to the epithelial cells of the endometrium, which compensatives by activating the PI3K/AKT/NF-κB signaling pathway and stimulating the production of glycosaminoglycan particles in uterine gland cells. Glycosaminoglycan particles are used to maintain pregnancy and protect the survival rate of fetal mice (Figure 6C).

## 4. Materials and Methods

### 4.1. Animal and Sample Collection

Eighty C57BL/6 J mice (female, 6 weeks old, 25 g body weight) were acquired from the Xi’an Jiaotong University (Xi’an, China). All experimental procedures in this study met the requirements of the Animal Ethics Committee of Northwest A&F University and complied with animal welfare provisions formulated by the Institutional Animal Care and Use Committee (18 February 2022).

Test 1: Twenty C57BL/6 J mice (female, 6 weeks old, 25 g body weight) were divided into two groups (n = 10 per group) and were treated with 0.01 M PBS buffer (pH 7.2) or LPS reagent (*Escherichia coli* [O55: B5]; Sigma, Burlington, MA, USA) by intragastric administration every other day for one month. MUC1, MUC2, MUC3, MUC4, MUC5ac, MUC5b, MUC6, MUC13, and MUC20 mRNA expressions in the uterine mucous layer were analyzed by qPCR, and MUC2 and MUC4 protein content of the uterine mucous layer was analyzed by ELISA.

Test 2: Sixty C57BL/6 J mice (female, 6 weeks old, 25 g body weight) were divided into two groups (n = 30 per group) and treated with 0.01 M PBS buffer (LPS (-) group) or LPS reagent (LPS (+) group, 1000 µg/kg) by intragastric administration every other day during the pregnancy period. The uterine mucous layer was collected on days 7 (n = 10) and 15 (n = 10) of pregnancy and on day 5 (n = 10) after LPS/PBS treatment. One part of the uterine sample was fixed in 10% formalin PAS staining and TEM. Total RNA and protein were extracted from another uterine sample for qPCR and WB and stored at −80 °C.

### 4.2. Periodic Acid–Schiff Stain (PAS Stain)

Mucopolysaccharide production by glandular cells in LPS (+) and LPS (-) groups was determined using periodic acid–Schiff (PAS) staining. Following LPS/PBS treatment, samples were taken during pregnancy and childbirth for histological examination by 10% formalin fixation, paraffin embedding, and sectioning. The following procedure was used: PAS fixative solution (75% ethanol solution) for 10 min; 1.0% periodic acid solution, oxidized in the dark for 5 min; one quick wash with distilled water; add Schiff’s reagent, seal and dye in a 37 °C incubator for 1 h; wash twice with sulfurous acid solution, for about 1 min each time; two uick washes with distilled water; hematoxylin staining of nuclei for 1 min; 0.5% hydrochloric acid alcohol solution color separation (5–10 s), then subjected to microscopic examination.

### 4.3. Immunohistochemistry

Confocal microscopy was used to assess the localization of MUC2 (Cat. No. ab90007; 5 µg; Abcam, Cambridge, UK) and MUC4 (Cat. No. Ab60720; 5 µg; Abcam) in the uterus on days 7 and 15 of pregnancy and day 5 after childbirth, after LPS/PBS treatment. All samples were fixed with 10% formalin for 1 day and washed 3 times with PBS. 0.3% hydrogen peroxide in methanol for 30 min, then 0.3% Triton × 100 (30% Triton × 100 + 0.01 MKPBS 100 mL) for 30 min. Tissues were incubated with anti-MUC2 and anti-MUC4 antibodies. AntiMUC2 (1:500) and antiMUC4 antibodies (1:200) were diluted in TBST (antibody dilution buffer) for 12 h at 4 °C and washed 3 times. Subsequently, tissues were incubated with Cy5-PEG-biotin mouse anti-rabbit IgG and FITC-PEG-biotin mouse anti-rabbit IgG (Bioss Antibodies, Woburn, MA, USA) at 37 °C for 2 h in the dark, and then stained with DAPI for 5 min. Finally, tissues were visualized under an LSM780 immunofluorescence microscope (Carl Zeiss, Inc., Thornwood, NY, USA). All experiments were repeated at least three times.

### 4.4. Transmission Electron Microscopy (TEM) Analysis

Mouse uteruses on days 7 and 15 of pregnancy and on day 5 after childbirth were analyzed by transmission electron microscopy (TEM). After fixation with 3% buffered glutaraldehyde, the tissue underwent the following procedures: (1) Rinsing: 0.1 M (pH 7.2) PBS buffer 4 times, for 10 min each time; (2) Post-fixation: fix with about 0.7 mL of 1% osmic acid at 4 °C for 1–2 h; (3) Post-rinse: 0.1 M (pH7.2) PBS buffer 3 times, for 10 min each time; (3) Dehydration: 30%, 50%, 70%, 80%, 90% ethanol solution once, for 15 min; 100% ethanol twice, for 10 min each time; (4) CO2 drying, sticking to the table, gold spraying, and sample observation.

### 4.5. qPCR

The total RNA was extracted from the uterus of female mice using Trizol reagent (Takara, Maebashi, Japan), according to the manufacturer’s protocol. First-strand cDNA synthesis from total RNA was performed using the cDNA synthesis kit (Bio-Rad, Hercules, CA, USA) according to the manufacturer’s instructions. UltraSYBR Mixture was used to PCR amplify 2µL of cDNA in 25 µL. The PCR mixture was denatured at 94 °C for 2 min, followed by 40 cycles at 94 °C for 30 s, 60 °C for 30 s and 72 °C for 45 s. Relative quantification was performed using the 2^−∆∆Ct^ method for the housekeeping GAPDH gene. The mean Ct values from the three technical replicates were used for each biological replicate. The primers used are listed in Table 1.

### 4.6. Western Blot (WB)

Total protein from the uterus was extracted using radioimmunoprecipitation assay (RIPA) buffer (in mmol/L) (20 mM Tris-HCl [pH 7.5], 150 mM NaCl, 1 mM Na_2_EDTA, 1 mM EGTA, 1% Triton X-100, 100 NaF, and 1 Na_3_VO_4_) with 1% phenylmethylsulfonyl fluoride (PMSF) for 5 min. The supernatants were collected, and protein concentrations were determined using a BCA protein assay kit (Thermo Fisher Scientific, Waltham, MA, USA). Proteins were electrophoresed on 10% sodium dodecyl sulfate-polyacrylamide gel electrophoresis (SDS-PAGE) and transferred onto polyvinylidene fluoride (PVDF) membranes. The membrane was then blocked with 5% skim milk at room temperature for 2 h and incubated with specific primary antibodies (1:1000 in PBS buffer) at 4 °C overnight. The primary antibodies included anti-E-cadherin (cat. no. ab197751; 5 µg; Abcam), anti-Claudin-1 (cat. no. ab129119; 5 µg; Abcam), anti-ZO1 (Zonula occludens, cat. no. ab276131; 5 µg; Abcam), anti-occludin (cat. no. Epr20992; 5 µg; Abcam), anti-JAMA (junctional adhesion molecule A; cat. no. 277080; 5 µg; Abcam), and anti-β-actin. For secondary antibody incubation, PVDF membranes were incubated with corresponding horseradish peroxidase (HRP)-conjugated secondary antibodies (1:10,000, Dako, Glostrup, Denmark) for 2 h at room temperature. Protein bands were digitally imaged for densitometric quantification using software (Eastman Kodak Company, Rochester, NY, USA).

### 4.7. Statistical Analysis

Statistical analyses were performed using GraphPad Prism 5, and the data are expressed as the mean ± SD. Statistical analyses were two pairs of tests, performed to determine the statistical significance between means. (* statistical difference between LPS (+) and LPS (-) groups in the same period (*p* < 0.05); ** statistical difference between LPS (+) and LPS (-) groups in the same period (*p* < 0.01)).

## 5. Conclusions

The above results indicate that LPS affects the uterus of mice, damages the structure of the uterine mucous layer in early pregnancy, and inhibits the PI3K/AKT/NF-κB signaling pathway. As a result, the uterine gland cells secrete less mucin, which leads to lower fetal survival in the uterus. In the middle of pregnancy, LPS continues to damage the tight junction barrier of uterine epithelial cells and translocates to the cytoplasm, which stimulates the secretion of many mucins and glycosaminoglycan particles by activating the PI3K/AKT/NF-κB signaling pathway in epithelial cells. The survival rate of fetal mice did not differ between the LPS (+) and LPS (-) groups, suggesting that the mucous layer and glycosaminoglycan particles of the uterine gland cells have a significant effect on maintaining the survival rate of fetal mice during pregnancy.

## Figures and Tables

**Figure 1 ijms-23-09972-f001:**
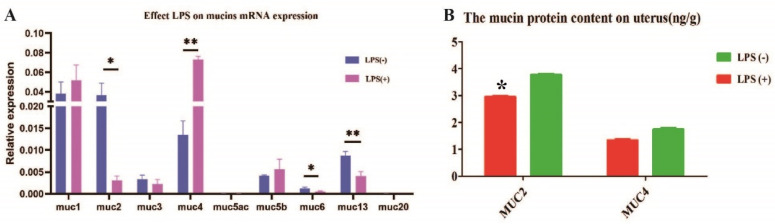
Effect of LPS on mucin mRNA and protein content of the endometrium in LPS (+) and LPS (-) groups. (**A**) Effect LPS on mucins mRNA expressions; (**B**) The mucin protein content on uterus. * Statistical difference between LPS (+) and LPS (-) groups in the same period (*p* < 0.05); ** statistical difference between LPS (+) and LPS (-) groups in the same period (*p* < 0.01).

**Figure 2 ijms-23-09972-f002:**
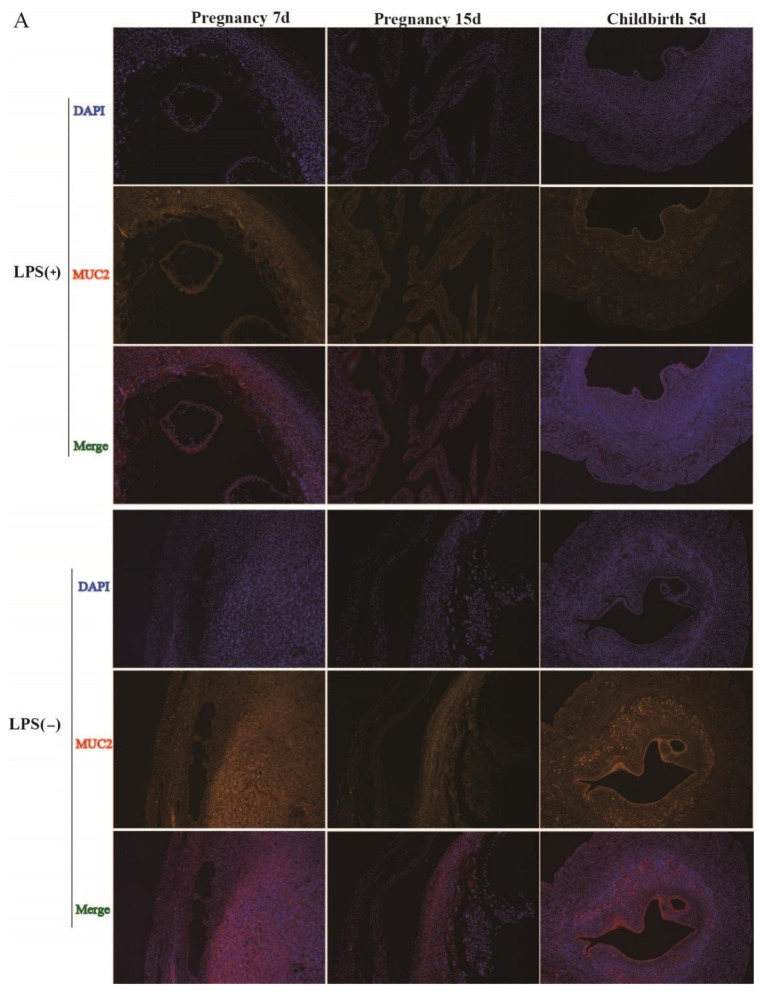
Effect of LPS on MUC2 and MUC4 expressions and distribution in uteri in LPS (+) and LPS (-) groups during pregnancy and after childbirth. Immunofluorescent staining results showed MUC2 (red) expression and nucleus (blue) in the uteri in LPS (+) and LPS (-) groups at day 7 and day 15 of pregnancy and 5 days after childbirth (×100) (**A**). Immunofluorescent staining results showed MUC4 (green) expression and nucleus (blue) in the uteri in LPS (+) and LPS (-) groups at day 7 and day 15 of pregnancy and 5 days after childbirth (×100) (**B**). ELISA results showed MUC2 and MUC4 protein content in endometria in LPS (+) and LPS (-) group at day 7 and day 15 of pregnancy and days after childbirth (**C**). * Statistical difference between LPS (+) and LPS (-) groups in the same period (*p* < 0.05).

**Figure 3 ijms-23-09972-f003:**
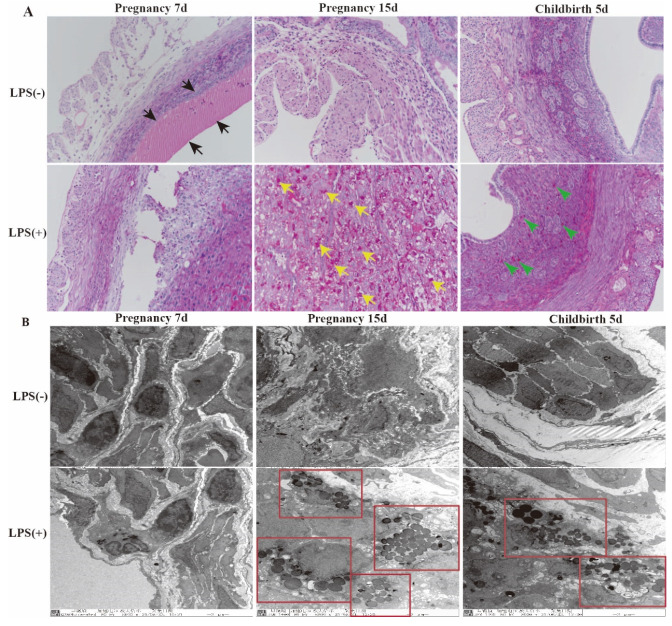
Effect of LPS on histomorphology and ultrastructure of uterus in mice. At days 7 of pregnancy, the thickness of the mucous layer of the uterus in mice in the LPS (-) group (black arrow). At day 15 of pregnancy, the endometrial glandular cells of the LPS (+) group secreted many glycosaminoglycan particles (yellow arrow). At 5 days after childbirth, the endometrial glandular cells of the LPS (+) group secreted few glycosaminoglycan particles (green arrow) (**A**) (×200). The ultrastructure of the uterus in mice at day 7 and day 15 of pregnancy and day 5 after childbirth (×8000). The endometrial glandular cells secreted more glycosaminoglycan particles at day 15 of pregnancy than at 5 days after childbirth in the LPS (+) group (red box) (**B**).

**Figure 4 ijms-23-09972-f004:**
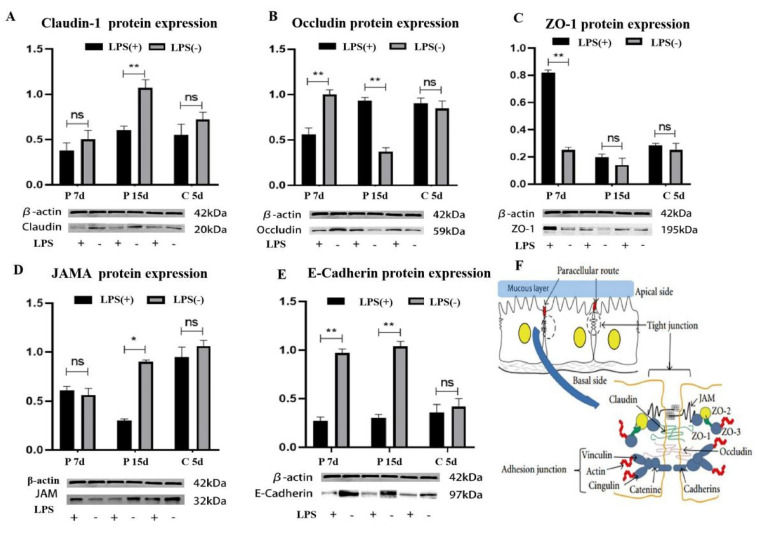
Effect of LPS on tight junction protein of endometrial epithelial cells at day 7 and day 15 of pregnancy and day 5 after childbirth. Claudin-1 protein expression in the uteri of female mice after LPS treatment (**A**). Occludin protein expression in the uteri of female mice after LPS treatment (**B**). ZO-1 protein expression in the uteri of female mice after LPS treatment (**C**). JAMA protein expression in the uteri of female mice after LPS treatment (**D**); E-cadherin protein expression in the uteri of female mice after LPS treatment (**E**); and the structure diagram of endometrial epithelial cells (**F**). * Statistical difference between LPS (+) and LPS (-) groups in the same period (*p* < 0.05); ** statistical difference between LPS (+) and LPS (-) groups in the same period (*p* < 0.01).

**Figure 5 ijms-23-09972-f005:**
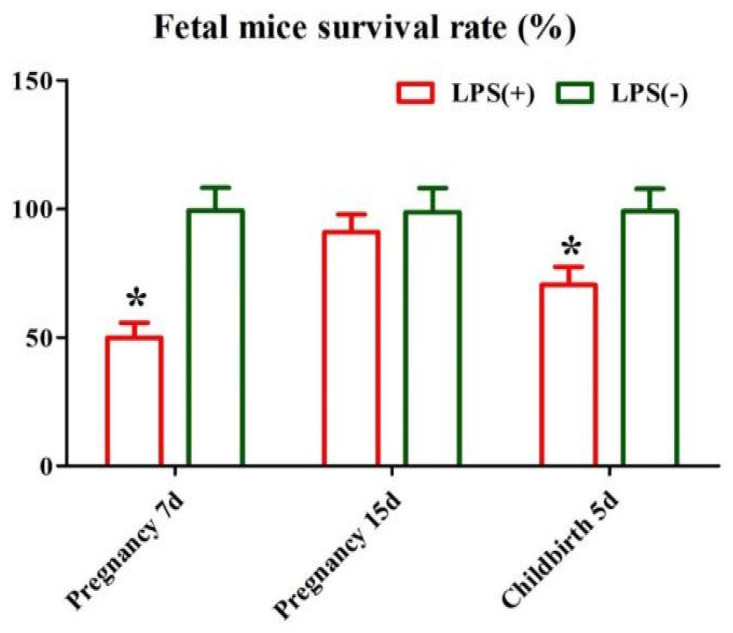
Effect of LPS on reproductive performance of female mice. * Statistical difference between LPS (+) and LPS (-) groups in the same period (*p* < 0.05)).

**Figure 6 ijms-23-09972-f006:**
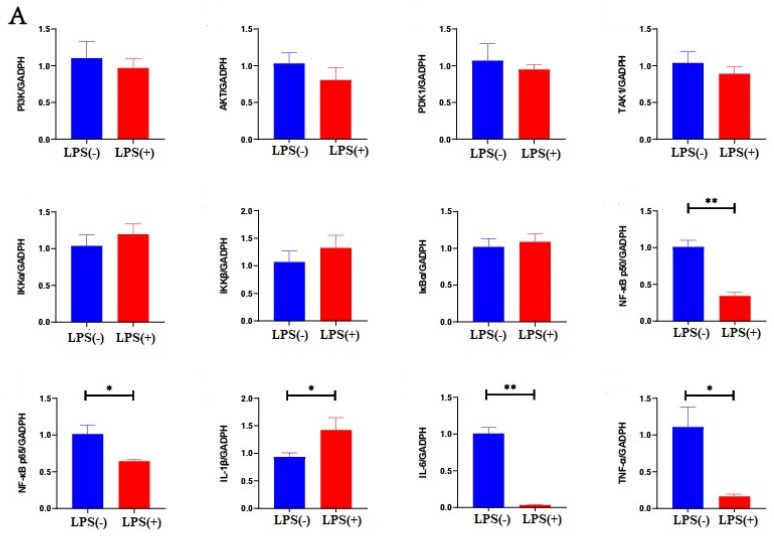
Effect of LPS on PI3K/AKT/NF-κB signaling pathway during pregnancy. At day 7 of pregnancy. Effect of LPS on factors influencing mRNA expression of PI3K/AKT/NF-κB signaling pathway in uteri of mice (**A**); Effect LPS on factors influencing mRNA expression of PI3K/AKT/NF-κB signaling pathway in uteri of mice at day 15 of pregnancy (**B**); Mechanism of LPS effect on PI3K/AKT/NF-κB signaling pathway in uteri of mice during pregnancy (**C**). * Statistical difference between LPS (+) and LPS (-) groups in the same period (*p* < 0.05); ** statistical difference between LPS (+) and LPS (-) groups in the same period (*p* < 0.01).

**Table 1 ijms-23-09972-t001:** Primer sequences of the genes.

Gene	Primer	Sequences	Product Size (bp)
PI3K	FP	GACCAATACTTGATGTGGCTGAC	190
	RP	TCCCTCGCAATAGGTTCTCC	
AKT	FP	CTCTAGGCATCCCTTCCTTACG	166
	RP	AGACACAATCTCCGCACCATAG	
PDK1	FP	TTTGAGACCATCACTTGGGAGA	174
	RP	CGTAGACAGGGAGTGGGAAGAG	
TAK1	FP	GCTCATTCATGGACATTGCTTC	133
	RP	CTCTGTTGCTTTGCCTGGTTT	
IKKα	FP	GCAGAGTCAGGACCGTGTT	192
	RP	CAGGCAATTTTAAGGAGGTG	
IKKβ	FP	TTCGCTACCCTTCCCCAATA	83
	RP	GGTGCCACATAAGCATCAGC	
IκBα	FP	ATGGAAGTCATTGGTCAGGTG	185
	RP	CAGGCAAGATGTAGAGGGGTAT	
NF-κB p50	FP	TCAAAATTTGCAACTATGTGGGG	166
	RP	AGGTTTGCAAAGCCAACCAC	
NF-κB p65	FP	AAGGACCTATGAGACCTTCAAGAG	115
	RP	GACAGAAGTTGAGTTTCGGGTAG	
IL-1β	FP	AAATCTCGCAGCAGCACAT	115
	RP	TCCTCATCCTGGAAGGTCC	
IL-6	FP	GACAAAGCCAGAGTCCTTCAGA	76
	RP	TGTGACTCCAGCTTATCTCTTGG	
TNF-α	FP	GATCGGTCCCCAAAGGGATG	92
	RP	CCACTTGGTGGTTTGTGAGTG	
GAPDH	FP	CCCTTAAGAGGGATGCTGCC	124
	RP	TACGGCCAAATCCGTTCACA	

## Data Availability

The study did not report any data.

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
