# Peer review of "The Mechanism of Lipopolysaccharide’s Effect on Secretion of Endometrial Mucins in Female Mice during Pregnancy"

_ijms, 2022, doi:10.3390/ijms23179972_

Round 1

Reviewer 1 Report (Previous Reviewer 1)

1.      Matching the color pattern and order (blue and pink) between RT PCR and ELISA for the LPS+ and LPS- groups in Figure 1 helps to avoid confusion for the reader.

2.      Dose not match the Muc2 and muc4 mRNA expression level and protein level

3.      A logical explanation is lacking whether there is a correlation between the expression levels of Muc2 and Muc4 and the change in thickness.

4.      The logical causal relationship between the expression levels of Muc2 and Muc4 and the expression of cell-to-cell connection proteins is unclear.

5.      It is recommended to add a size bar for the tissue section and provide a quantification of how much its thickness has changed in figure 2 (“146”) and how many particles are decreased in the LPS treatment group (“163”)

6.      Figure 4, it is rat or mice?

7.      Missing Figure 5 ‘s legend

8.      Wrong figure order ‘Figure 5’

9.       Figure 6 is not a good explanatory drawing of the draft as to why LPS application can reduce the thickness and mucin production. It looks like a general inflammatory mechanism.

Author Response

NO.: ijms-1879023

Title: The mechanism of Lipopolysaccharide effect on secretion of endometrial mucins in female mice during pregnancy

Corresponding Author: Mr. Chenchen Wu

Dear Editors and Reviewers:

   Thank you for your letter and for the reviewers’ comments concerning our manuscript entitled “The mechanism of Lipopolysaccharide effect on secretion of endometrial mucins in female mice during pregnancy”. Now I had fully evaluated and utilised and submitted my revised manuscript and ARRIVE checklist as an additional file. We have revised portion are marked in red in the paper conform to the journal style. Thanks again! The main corrections in the paper and the responds to the reviewer’s comments are as flowing:

Reviewer 1#

1.Matching the color pattern and order (blue and pink) between RT PCR and ELISA for the LPS+ and LPS- groups in Figure 1 helps to avoid confusion for the reader.

Answer: I had changed the colors in the picture to avoid confusion for the reader.

2.Dose not match the Muc2 and muc4 mRNA expression level and protein level

Answer: To confirm the effect of LPS on mucin (MUC1, MUC2, MUC3, MUC4, MUC5ac, MUC5b, MUC6, MUC13, and MUC20) levels in the endometrium, 20 female mice were intragastrically administered LPS reagent every other day for one month. The MUC2 and MUC4 mRNA expression were higher than other mucins in the endometrium, they were significantly difference between LPS(+) and LPS(-) group, therefore, we selected the MUC2 and MUC4 as the main components of mucus layer for analysis. I added the section on line 243-246 page 9.

  1. A logical explanation is lacking whether there is a correlation between the expression levels of Muc2 and Muc4 and the change in thickness.

Answer: Thanks for your advise. I had explained correlation between the expression levels of Muc2 and Muc4 and the change in thickness on line 243-247 page 9. Mucin (MUC)-2 and MUC4 are a secreted protein produced by epithelial goblet cells as the main component of mucus in the endometrium. In addition, I added the MUC2 and MUC4 protein expression in uterus at pregnancy 7d, 15d and childbirth 5d by  Immunofluorescent in figure-2.

  1. The logical causal relationship between the expression levels of Muc2 and Muc4 and the expression of cell-to-cell connection proteins is unclear.

Answer: LPS can increase the thickness of endometrial mucous layer, it do not damage cell-cell tight junctions due to the protection of the mucus layer in early pregnancy; LPS reduces the thickness of endometrial mucus layer, and the expression of tight junction protein significantly decreases, which destroys the cell-cell tight junctions in middle and last pregnancy. The above sentence has added in the paper on line 271-272 and line 279-283 page 10.

  1. It is recommended to add a size bar for the tissue section and provide a quantification of how much its thickness has changed in figure 2 (“146”) and how many particles are decreased in the LPS treatment group (“163”)

Answer: I added a size bar for the tissue section on line 352 page 12. I am very sorry.I did not record the number of particles in figue-2B.

6.Figure 4, it is rat or mice?

Answer: it is female mice.

  1. Missing Figure 5 ‘s legend

Answer: I am very sorry. I had revised in the paper.

  1. Wrong figure order ‘Figure 5’

 Answer: I am very sorry. I had revised in the paper.

  1. Figure 6 is not a good explanatory drawing of the draft as to why LPS application can reduce the thickness and mucin production. It looks like a general inflammatory mechanism.

Answer: I have redrawn the Figure-6C.

Reviewer 2 Report (New Reviewer)

This is my review on mechanism of Lipopolysaccharide effect on secretion of endometrial mucins in female mice during pregnancy.

The authors try to elucidate significant aspects of reproductive disorders; endotoxins. The aim was to evaluate LPS and the mucous layer of the endometrium. Introduction should include more information regarding the endotoxins, their sources and pathophysiology.

Materials and methods are adequately described.

Results are well presented. Figure 2 should contain on each image a scale bar. Figure 4 should be improved in terms of quality. 

This is an interesting study with important conclusions. 

Language is fine.

Author Response

NO.: ijms-1879023

Title: The mechanism of Lipopolysaccharide effect on secretion of endometrial mucins in female mice during pregnancy

Corresponding Author: Mr. Chenchen Wu

Dear Editors and Reviewers:

   Thank you for your letter and for the reviewers’ comments concerning our manuscript entitled “The mechanism of Lipopolysaccharide effect on secretion of endometrial mucins in female mice during pregnancy”. Now I had fully evaluated and utilised and submitted my revised manuscript and ARRIVE checklist as an additional file. We have revised portion are marked in red in the paper conform to the journal style. Thanks again! The main corrections in the paper and the responds to the reviewer’s comments are as flowing:

Reviewer 1#

This is my review on mechanism of Lipopolysaccharide effect on secretion of endometrial mucins in female mice during pregnancy.The authors try to elucidate significant aspects of reproductive disorders; endotoxins. The aim was to evaluate LPS and the mucous layer of the endometrium. Introduction should include more information regarding the endotoxins, their sources and pathophysiology.

Materials and methods are adequately described.

Results are well presented. Figure 2 should contain on each image a scale bar. Figure 4 should be improved in terms of quality. 

This is an interesting study with important conclusions. 

Language is fine.

Answer: Thank you very much! I had revised the Figure-2 and Figure-4 in this paper. 

This manuscript is a resubmission of an earlier submission. The following is a list of the peer review reports and author responses from that submission.

Round 1

Reviewer 1 Report

The manuscript describes genetic changes in cell lines after LPS application using a transwell system.   The authors suggest that the ECM receptor change after LPS application is a co-culture system that is important to the endothelial barrier mechanism. However, this manuscript needs correction of the errors presented below.

Figure 1,

With Figure 1, it is difficult to find sufficient evidence that 1:3 is the best system. (N number missing)

Since the authors compared two types of systems, it is unreasonable to claim that they are optimal.
 The authors did not describe how repeatedly they tested and did not perform statistical analysis, so it is difficult to determine whether this is a meaningful result.

Figure 1C: The quality of the image is not enough to determine whether a 1:3 ratio is a better system, so it is necessary to present a better picture.

Figure 2: 

N number missing, no statistical analysis
Picture 2D quality is poor, so it's hard to distinguish the difference.

Figure 3:

N number missing, A, B: no statistical analysis, C: T-test?  Unclear or wrong statistical analysis (One Way ANOVA). 

Figure 3D: Without a control or DAPI stain and scramble transfection control image, it is difficult to distinguish between signal or noise.

Figure 4,5: N number missing. 

Reviewer 2 Report

Dear authors,

Thank you for your interesting work. Although it is a well designed project, I have few suggestions that need your attention in order to achieve a publication of your work. 

I would suggest you to undertake few more experiments and employ a control group, LPS (-), in the 3.5 section.  

You should add references throughout the manuscript, specify details of all instruments used in the experiments, and name all the reagents and the the companies obtained from.

lines 54-55: please clarify

84: define CCK-8 reagent?

188-189: should ne in the discussion section

190: ?? 1

312:  clarify

438-439: rephrase-clarify

447: rephrase